# *Prevotella copri* alleviates hyperglycemia and regulates gut microbiota and metabolic profiles in mice

Caixin Yang,[1,2,3] Ruiting Lan,[4] Lijun Zhao,[1,2] Ji Pu,[2] Dalong Hu,[4] Jing Yang,[2,5,6] Huimin Zhou,[1,2] Lichao Han,[2] Lin Ye,[1,2] Dong Jin,[2,5,6] Jianguo Xu,[1,2,3,5,7] Liyun Liu[1,2,5,6]

**ABSTRACT** *Prevotella copri* is the dominant species of the *Prevotella* genus in the gut, which is genomically heterogeneous and difficult to isolate; hence, scarce research was carried out for this species. This study aimed to investigate the effect of *P. copri* on hyperglycemia. Thirty-nine strains were isolated from healthy individuals, and three strains (HF2123, HF1478, and HF2130) that had the highest glucose consumption were selected to evaluate the effects of *P. copri* supplementation on hyperglycemia. Microbiomics and non-target metabolomics were used to uncover the underlying mechanisms. Oral administration of *P. copri* in diabetic db/db mice increased the expression and secretion of glucagon-like peptide-1 (GLP-1), significantly improved hyperglycemia, insulin resistance, and lipid accumulation, and alleviated the pathological morphology in the pancreas, liver, and colon. *P. copri* changed the composition of the gut microbiota of diabetic db/db mice, which was characterized by increasing the ratio of Bacteroidetes to Firmicutes and increasing the relative abundance of genera *Bacteroides*, *Akkermansia*, and *Faecalibacterium*. After intervention with *P. copri*, fecal metabolic profiling showed that fumaric acid and homocysteine contents decreased, and glutamine contents increased. Furthermore, amino acid metabolism and cAMP/PKA signaling pathways were enriched. Our findings indicate that *P. copri* improved glucose metabolism abnormalities in diabetic db/db mice. Especially, one of the *P. copri* strains, HF2130, has shown superior performance in improving hyperglycemia, which may have the potential as a probiotic against hyperglycemia.

**IMPORTANCE** As a core member of the human intestinal ecosystem, *Prevotelal copri* has been associated with glucose metabolic homeostasis in previous studies. However, these results have often been derived from metagenomic studies, and the experimental studies have been based solely on the type of strain DSM 18205[T]. Therefore, more experimental evidence from additional isolates is needed to validate the results according to their high genomic heterogeneity. In this study, we isolated different branches of strains and demonstrated that *P. copri* could improve the metabolic profile of hyperglycemic mice by modulating microbial activity. This finding supports the causal contribution of *P. copri* in host glucose metabolism.

**KEYWORDS** *Prevotella copri*, hyperglycemia, GLP-1, gut microbiota, metabolism

Address correspondence to Liyun Liu, liuliyun@icdc.cn, or Jianguo Xu, xujianguo@icdc.cn.

The authors declare no conflict of interest.

See the funding table on p. 16.

The *Prevotella*-dominated gut microbiome is one of the three gut types of the human gut microbiome (1). In recent years, several metagenomic studies have demonstrated that *P. copri* is more abundant in non-Western populations and have linked high-fiber dietary patterns with a high abundance of *P. copri* (2, 3). In 2020, our laboratory analyzed gut microbiota from 120 Chinese healthy human people at the "species" level using bacterial macrotaxonomy and found *P. copri* to be a representative species of the *Prevotella* enterotype (4). *P. copri* is strictly anaerobic and difficult to isolate

and culture; thus, previous studies have focused on macrogenomic analysis, and the physiological and metabolic properties of the strains are only based on the type of strain DSM18205$^T$. In 2019, Tett et al. reconstructed more than 1,000 *P. copri* genomes by means of metagenome assembly, demonstrating that *P. copri* has four independent branches with high genomic heterogeneity (5). Whether genomic diversity in *P. copri* is linked to functional diversity has not been fully explored.

With the development of metagenomic and metabolomic analyses of the correlation between gut microbes and host health, it is found that hosts with higher *P. copri* abundance were associated with lower c-peptide, triglyceride, and postprandial blood glucose levels, which is helpful in reducing hyperglycemia (6). Type two diabetes mellitus (T2DM) is a metabolic disease characterized by hyperglycemia, which may lead to complications and serious disease burden (7). T2DM is primarily caused by defective insulin secretion by pancreatic $\beta$-cells and reduced sensitivity of the body to insulin (8). Enteric glucagon-like peptide-1 (GLP-1), secreted by enteroendocrine L cells, can stimulate insulin secretion by $\beta$-cells and improve responsiveness to glucose (9). It has been reported that some species of *Lactobacillus* have been shown to enhance insulin secretion through glucose-triggered GLP-1 secretion to reduce hyperglycemia (10, 11). However, strain specificity needs to be considered in assessing the effect of gut microbes on human health or disease.

In this study, we isolated and selected three *P. copri* strains and investigated the intervention effect of *P. copri* on the db/db mice by measuring the glucose index. Furthermore, we combined the analysis of microbiome and metabolomics profiling to explore the possible mechanisms of hyperglycemia. The results showed that *P. copri* could alleviate hyperglycemia by regulating the gut microbiota and metabolites and activating the intestinal cAMP/PKA signaling pathway to promote the GLP-1 release.

## RESULTS

### Phylogenetic analysis and glucose consumption determination of 39 *P. copri* strains

A total of 39 *P. copri* strains were isolated from 27 healthy human fecal samples and sequenced. Phylogenomic analysis of these 39 strains together with 145 publicly available *P. copri* genomes found that the 39 strains in this study belonged to clades A (32/39) and C (7/39) of the clades (A-D) previously defined by *Tett* et al. (5), which was consistent with the average nucleotide identity (ANI) values (Fig. S1B and C). The ANI values were greater than 95% for strains within clades and less than 90% between clade groups. We further classified the 32 clades A strains from this study into sub-clade A1 (22/32) and A2 (10/32) (Fig. S1E and F).

To select *P. copri* strains for further study, we examined the glucose consumption of 39 *P. copri* strains in RPMI-1640 medium. As shown in Fig. S2, the 39 *P. copri* strains had differences in glucose consumption. Strains HF2123, HF1478, and HF2130 in clades A1, A2, and C, respectively, had the highest glucose consumption and were selected.

To check the antibiotic-resistant genes and potential virulence genes carried by *P. copri* strains HF2123, HF1478, and HF2130, we used AMRFinderPlus to screen for antibiotic resistance genes and found that strain HF2123 carried *CfxA6* and *ErmG*, strain HF1478 carried *CfxA6*, *CfxA, Inu(AN2)*, and *Mef(En2)*, and strain HF2130 carried *CfxA6* and *ErmG*. We also tested the three *P. copri* strains for resistance to penicillin, ampicillins clindamycin, clarithromycin, erythromycin, and azithromycin phenotypically. These three strains were resistant to penicillin G and ampicillin, but sensitive to clindamycin, clarithromycin, erythromycin, and azithromycin. We used the virulence factor database (VFDB) to predict virulence genes with an identity threshold of 80% and found the three strains carried no known virulence genes.

## P. copri HF2123, HF1478, and HF2130 improved blood glucose levels and insulin resistance of db/db mice

To observe the changes in blood glucose levels in db/db mice during 4 weeks of continued interventions with *P. copri* HF2123, HF1478, and HF2130, we tested fasting blood glucose (FBG) once a week and performed an oral glucose tolerance test (OGTT) in the final week. FBG of the diabetes group was significantly higher than that of the metformin and HF2130 intervention groups from the second week onward and that of the HF2123 and HF1478 intervention groups from the third week onward ($P < 0.05$) (Fig. 1A). The OGTT results showed that the blood glucose levels in the diabetes group were significantly higher than those in the other groups at each time point of measurements

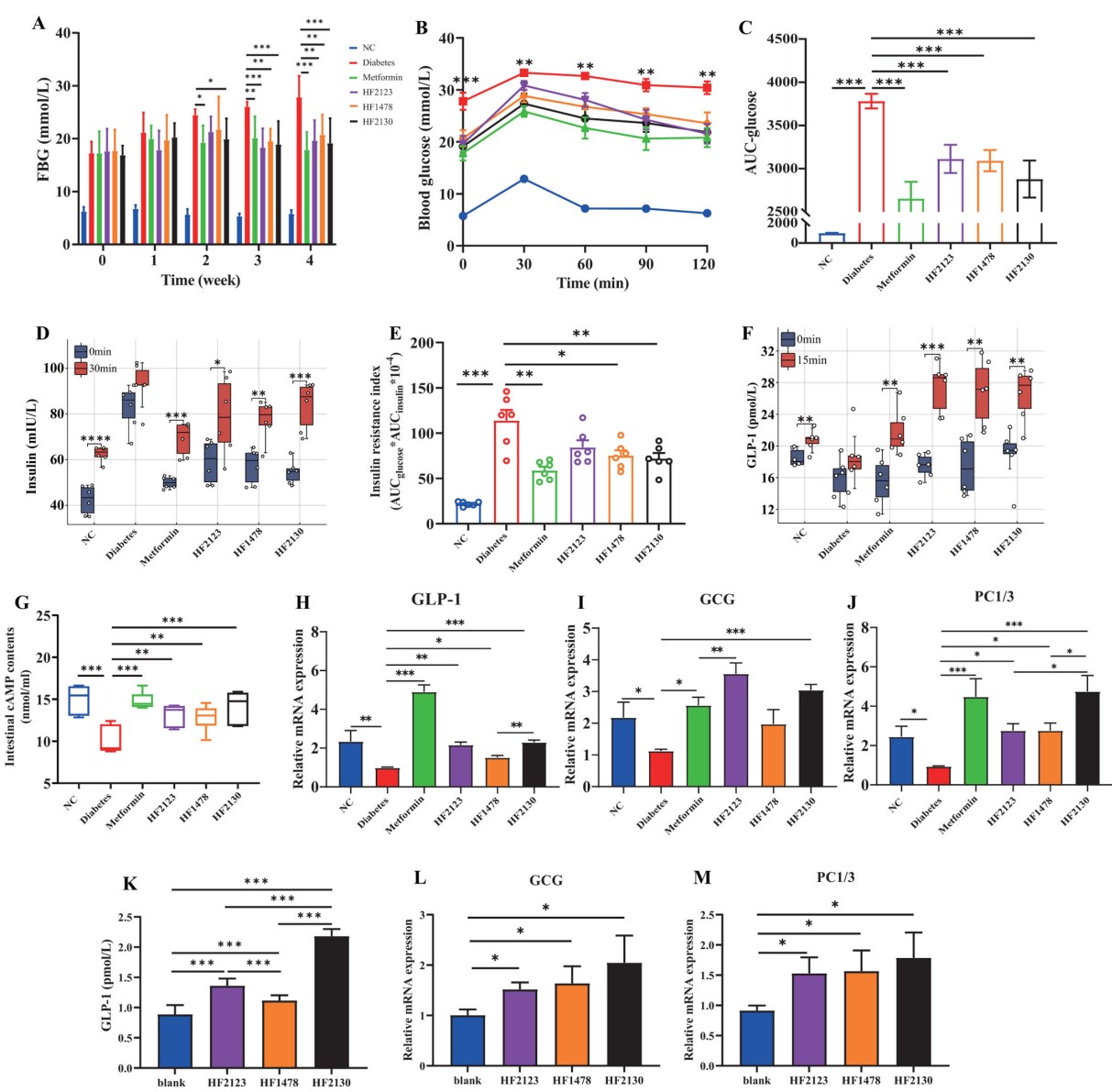

**FIG 1** *P. copri* HF2123, HF1478, and HF2130 improved blood glucose levels and insulin resistance through promoting GLP-1 of db/db mice. (A) The levels of FBG, (B) the curve of OGTT, (C) the AUC based on the OGTT, (D) insulin levels in the plasma, (E) insulin resistance index, (F) GLP-1 levels in the plasma, (G) intestinal cAMP levels, (H–J) the mRNA levels of GLP-1, GCG, and PC1/3 in the colon tissue were determined by qRT-PCR, (K) GLP-1 levels in supernatant of NCI-H716 cells, and (L–M) the mRNA levels of GCG and PC1/3 in NCI-H716 cells. Values are presented as mean ± SEM with each group ($n = 6$). *$P < 0.05$, **$P < 0.01$, ***$P < 0.001$. FBG, Fasting blood glucose; OGTT, Oral glucose tolerance; AUC, Area under the curve; GLP-1, glucagon-like peptide-1; GCG, Proglucagon; PC1/3, Proconvertase 1/3.

(0, 30, 60, 90, and 120 min after oral glucose) (Fig. 1B). Furthermore, by AUC, the blood glucose levels in all three *P. copri* treatment groups were significantly lower in the diabetes group (Fig. 1C). We then measured the plasma insulin levels at fasting and 30 min after glucose feeding. Compared with the diabetes group, fasting insulin was significantly decreased in all three *P. copri* treatment groups ($P < 0.05$). After 30 min, the elevation of insulin in the *P. copri* treatment groups was significantly higher than that in the diabetes group (Fig. 1D). Calculation of the insulin resistance index showed that *P. copri* treatment alleviated insulin resistance in the db/db mice (Fig. 1E).

To explore whether *P. copri* stimulated insulin release by increasing GLP-1 secretion, plasma GLP-1 was examined in all groups. The GLP-1 level of the three *P. copri* treatment groups significantly increased after 15 min of oral glucose ($P < 0.05$), whereas the increase of GLP-1 in the diabetes group was not statistically significant (Fig. 1F). After 4-week intervention, we determined the intestinal cAMP levels that drive the insulino-tropic effects of GLP-1, as well as the mRNA levels of GLP-1-related genes proglucagon (*GCG*) and Proconvertase 1/3 (*PC1/3*). Compared with diabetes group, the cAMP level was significantly higher in the *P. copri* treatment groups (Fig. 1G). In addition, the mRNA expression levels of *glp-1*, *GCG*, and *PC1/3* in HF2123 and HF2130 groups were significantly higher than those in the diabetes group ($P < 0.05$). However, the mRNA expression levels of *GCG* were comparable between the HF1478 group and the diabetes group ($P > 0.05$). The treatment by HF2130 exerted a stronger effect on the expression of *PC1/3* in db/db mice than by HF2123 and HF1478 ($P < 0.05$) (Fig. 1H through J). Furthermore, we assessed the secretion of GLP-1 and the expression of related genes in NCI-H716 cells. Consistent with the results of animal experiments, NCI-H716 cells treated with *P. copri* strains HF2123, HF1478, and HF2130 had increased secretion of GLP-1 and the expression of *GCG* and *PC1/3* genes (Fig. 1K through M). Notably, the increase of GLP-1 in the HF2130 treatment group was twice that of the other groups ($P < 0.001$) (Fig. 1K). Thus, *P. copri* treatment elevated the cAMP levels in the intestine to promote GLP-1 secretion and thereby stimulate insulin secretion.

## *P. copri* altered lipid profile in db/db mice

Hyperlipidemia is a complication of hyperglycemia, and elevated lipids predispose to cardiovascular disease. Compared with the NC group, serum TC and TG concentrations in diabetes group were significantly increased, whereas these serum lipid profile deviations were averted in *P. copri* HF2123, HF1478, and HF2130 treatment groups ($P < 0.05$, Fig. S3A through D). The serum high-density lipoprotein cholesterol (HDL-C) and low-density lipoprotein cholesterol (LDL-C) concentrations were comparable between the NC and the diabetes groups ($P > 0.05$). Additionally, after 4 weeks of intervention, the weight gain of diabetes group was significantly higher than that of the other groups ($P < 0.01$, Fig. S3E).

## *P. copri* prevented histological damage to the pancreas, liver, and colon in db/db mice

For further assessment of the effect of *P. copri* on histopathology of the colon, liver, and pancreas in db/db mice, hematoxylin-eosin staining (HE) was performed, and the results are shown in Fig. 2. Pancreatic tissues in the diabetes group showed islet cell necrosis, lymphocyte infiltration, and loosely arranged islet cells with a decreased islet area. In contrast, the HF2123 treatment group had a clear structure of the layers and a significantly larger size of the pancreatic islets. HF1478 and HF2130 treatment groups still had small lymphocytic infiltration. Compared with the diabetes group, hepatic tissues in the metformin and *P. copri* treatment groups showed different degrees of attenuation of hepatocyte hydropic degeneration, diffuse ballooning degeneration, diffuse steatosis, and cytoplasmic vacuolization. Diabetes group was observed to have small areas of intestinal tissue with erosion, necrosis of the intestinal epithelium, necrosis of the lamina propria intestinal glands with loss of structure, hyperplasia of connective tissue with a small amount of lymphocytic infiltration, and atrophy of the muscular layer with reduced

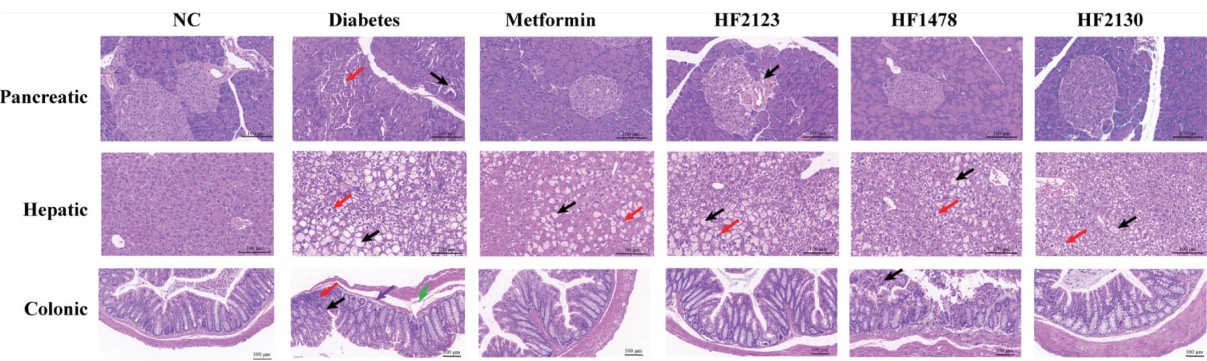

**FIG 2** Effects of *P. copri* on the pancreas, liver, and colon tissues. Hematoxylin-eosin (HE) stained pancreatic [40×; acinar necrosis with lymphocyte infiltration (black arrow); loose arrangement of islet cells (red arrow)], hepatic [40×; diffuse steatosis (black arrow); hepatocyte necrosis with cytoplasmic vacuolization (red arrow)], and colonic [20×; intestinal epithelial cells and intestinal glands are necrotic (black arrows), necrosis of intestinal glands in lamina propria, (red arrows); lymphocytic infiltration (purple arrows); and edema of submucosa, (green arrows)].

thickness. Compared with diabetes group, all other groups showed apparent repair and reduction in lymphatic infiltration.

## *P. copri* supplementation modulated gut microbiota in db/db mice

To investigate the abundance of *P. copri* in feces of db/db mice after 4 weeks of continuous gavaging of *P. copri* in the HF2123, HF1478, and HF2130 treatment groups, *P. copri* species-specific quantitative PCR (qPCR) was performed to detect the absolute abundance of *P. copri* before and after intervention. As shown in Fig. 3A, the abundance of *P. copri* was significantly increased in the three *P. copri* treatment groups after 4 weeks of intervention. Notably, compared with the start of the experiment, the abundance of *P. copri* was markedly decreased in diabetes group after 4 weeks of the experiment (Fig. 3A).

Based on the 16S rRNA gene amplicon sequencing result, at the phylum level, Firmicutes dominated the gut microbiota in the diabetes group, whereas Bacteroidetes dominated the gut microbiota in other groups (Fig. 3B). The ratio of Bacteroidetes to Firmicutes in diabetes group was lower than that in other groups ($P < 0.05$, Fig. 3C). We further analyzed the relative abundance of samples in each group from phylum to genus level by linear discriminant analysis effect size (LEfSe). The relative abundance of the genus *Pseudomonas*, *Lachnospiraceae* NK4A136 group, and *Mucispirillum* was increased in the diabetes group compared with the *P. copri* treatment groups. In contrast, the relative abundance of the genera *Prevotella*, *Bacteriodes*, and *Faecalibacterium* increased in the HF2123 and HF2130 treatment groups compared with the diabetes group. Additionally, the relative abundance of the genera *Prevotella* and *Akkermansia* increased in the HF1478 treatment group (Fig. 3D through G). We also assessed differences in some critical genera between the diabetes group and the *P. copri* treatment groups. *Lachnospiraceae* NK4A136 ($P = 0.01–0.09$) was significantly increased, whereas *Bacteriodes* ($P = 0.002–0.042$) were significantly decreased in the diabetes group. *Prevotella* ($P = 0.006–0.038$) increased in all three *P. copri* treatment groups. In addition, *Faecalibacterium* ($P = 0.001–0.002$) was more abundant in the HF2123 and HF2130 treatment groups. *Akkermansia* ($P = 0.03$) was significantly higher in the HF1478 treatment group (Fig. 3H). No differences were observed for *Alistipes*, *Bifidobacterium*, and *Lactobacillus* among the groups ($P > 0.05$).

## *P. copri* treatment altered fecal metabolic profiles in db/db mice

Metabolic profiles are associated with the severity of metabolic diseases. The fecal metabolites were identified using liquid chromatography-tandem mass spectrometry (LC-MS/MS). The relationship between metabolite expression and different groups was

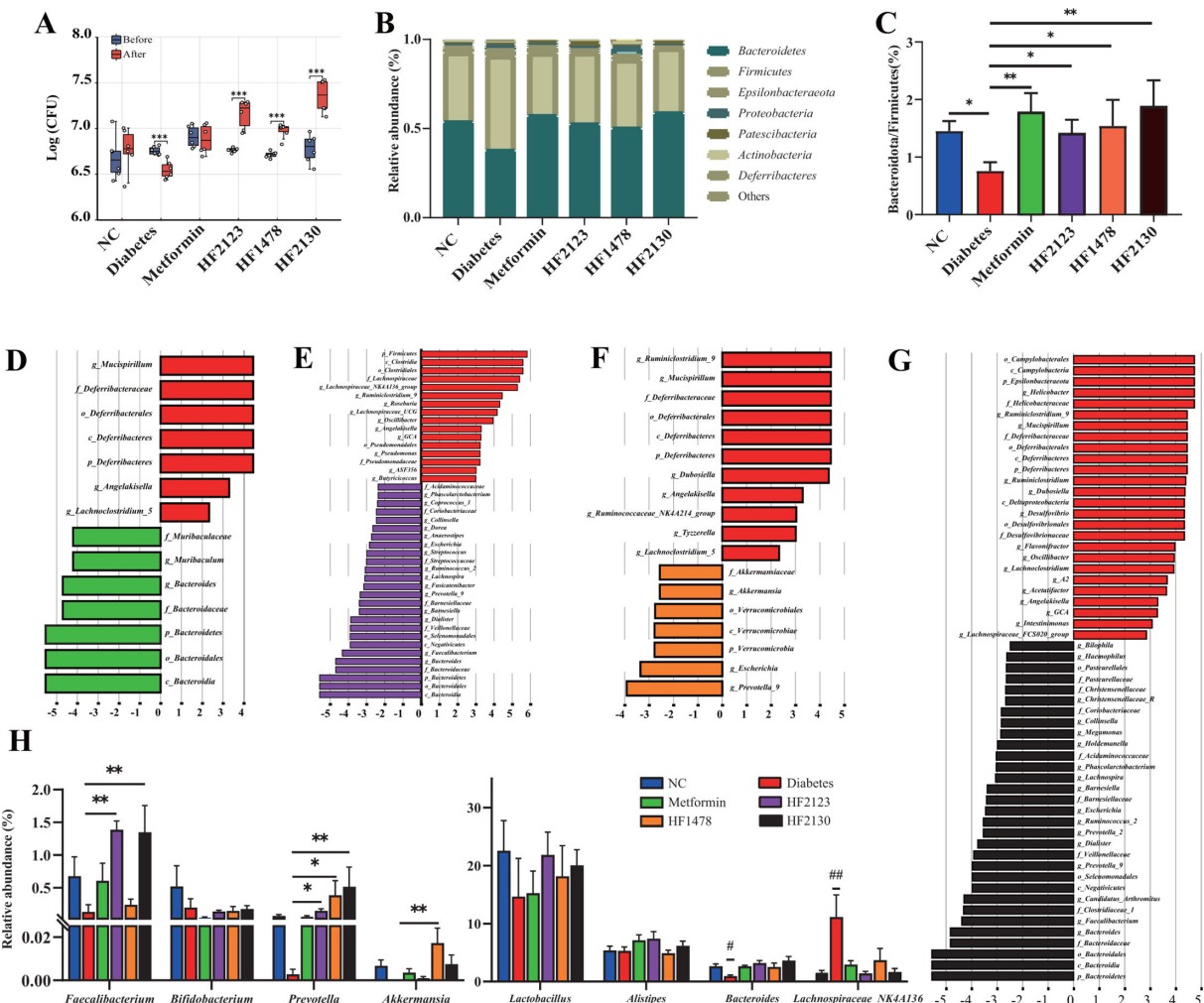

**FIG 3** Effects of the *P. copri* on gut microbiota. (A) qRT-PCR for *P. copri* levels in feces before and after intervention (*n* = 6). (B) Composition of major bacterial communities at the phylum level. (C) Ratio of Bacteroidota to Firmicutes. (D–G) Discriminative taxa from phylum to genus level as determined by linear discriminant analysis (LDA) effect size. (H) Relative abundance of *Lactobacillus*, *Alistipes*, *Akkermansia*, *Bacteroides*, *Prevotella*, *Faecalibacterium*, *Bifidobacterium*, and *Lachnospiraceae* NK4A136. Data are expressed as the mean ± SEM. *$P < 0.05$, **$P < 0.01$ vs diabetes group. #, $P < 0.05$, ##, $P < 0.01$ vs other groups.

modeled using the Partial Least Squares Discriminant Analysis (PLS-DA) method (Fig. 4A through D). The R2Y and Q2Y values of the metformin group vs the diabetes group and the HF2130 treatment group vs the diabetes group exceeded 0.4, indicating that these models effectively explained and predicted the data, whereas the R2Y values of the HF2123 treatment group and the HF1478 treatment group vs the diabetes group were close to 1.0, indicating that the model had a better explanatory rate but limited predictive ability. As detailed in Table S3, 380 (metformin group), 38 (HF2123 treatment group), 96 (HF1478 treatment group), and 121 (HF2130 treatment group) metabolites were significantly changed (VIP >1.0 and $P < 0.05$) compared with the diabetes group, respectively (Table S3). The downregulated metabolites included fumaric acid, bilirubin, and homocysteine, and the upregulated metabolites included arachidonic acid, glutamine, and critic acid.

Differential metabolites were analyzed for the enrichment of metabolic pathways by the Kyoto Encyclopedia of Genes and Genomes (KEGG) database. Alanine, aspartate, and glutamate metabolisms were enriched in the metformin group. Taurine and hypotaurine metabolism were enriched in the metformin group and HF2123 treatment group. Leucine, isoleucine, and valine degradation pathways and butanoate metabolism were

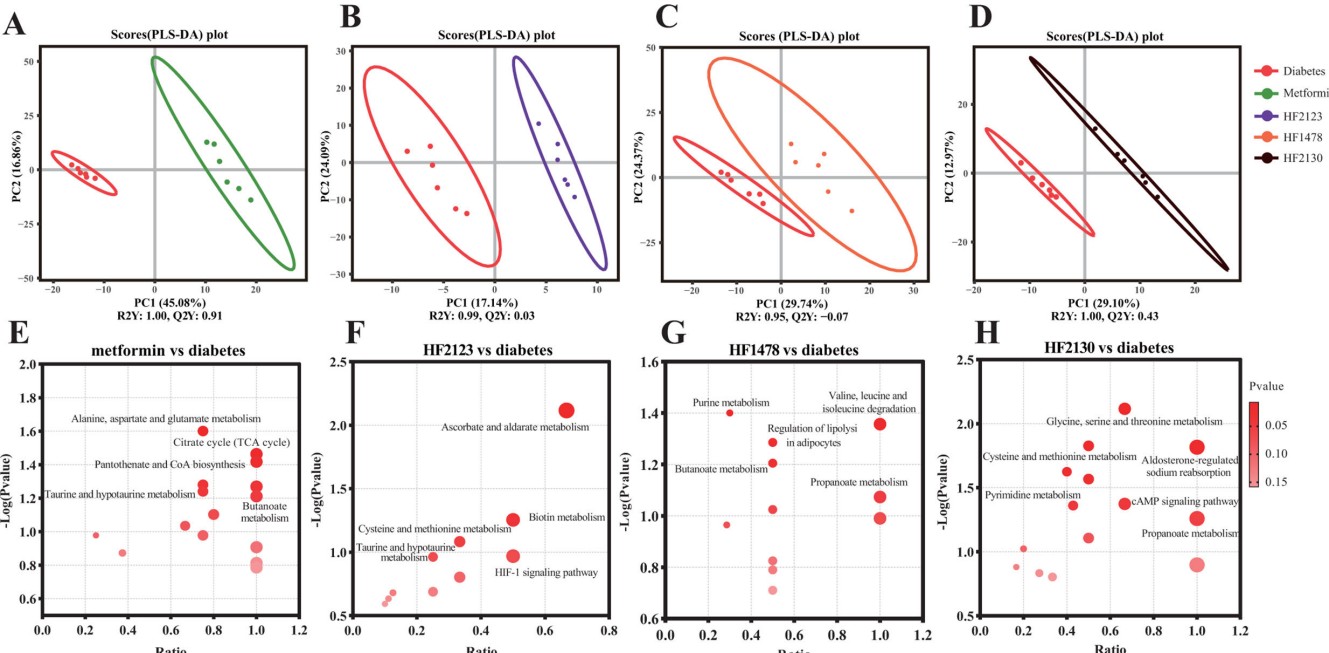

**FIG 4** Effect of *P. copri* on fecal metabolic profiles of db/db mice. (A–D) PLS-DA analysis of groups metformin, HF2123, HF1478, HF2130, and diabetes group, respectively. (E–H) KEGG topological analysis of groups metformin, HF2123, HF1478, HF2130, and diabetes group, respectively. Bubble size represents the ratio, and color represents the *P*-value. The larger the bubble, the darker the red color, the more important the pathway.

enriched in the HF1478 treatment group. The cAMP pathway and glycine, serine, and threonine metabolisms were enriched in the HF2130 treatment group (Fig. 4E through H).

## *P. copri* modulated the intestinal cAMP/PKA pathway

Based on the above metabolomics results, we validated the pathway differences by examining the phosphorylation levels of the downstream proteins PKA and CREB of the cAMP pathway. Relative quantitative analysis of p-PKA and p-CREB protein levels in mouse intestinal tissues showed that PKA and CREB phosphorylation levels were significantly increased in HF2123, HF1478, and HF2130 treatment groups than those in the diabetes group (Fig. 5A). This result indicated that all three *P. copri* strains HF2123, HF1478, and HF2130 could activate the intestinal cAMP/PKA signaling pathway in db/db mice.

In addition, we observed pathological changes in pancreatic islet cells in HE. To explore the cause, we examined the levels of the anti-apoptosis protein bcl-2 and the caspase-3 protein, which can induce apoptosis in the pancreas. Compared with the diabetes group, *P. copri* HF2123, HF1478, and HF2130 treatment significantly decreased caspase-3 protein and increased bcl-2 protein levels in pancreatic tissues of db/db mice (Fig. 5B). The results indicated that *P. copri* may protect pancreatic *β*-cells from apoptosis by upregulating the bcl-2 level.

## Correlation analysis of gut microbiota and metabolic profiles

Spearman's correlation analysis was used to evaluate the relationship between differential microbiota and metabolic profiles, as well as physiological and biochemical indices such as fasting blood glucose, area under the oral glucose tolerance curve, insulin increment, GLP-1 increment, total cholesterol, and triglycerides, after intervention with *P. copri* in db/db mice. As shown in Fig. 6, There was a significant negative correlation between *Prevotella* and *Bacteroides* with the level of fumaric acid ($P < 0.05$ or $P < 0.01$), whereas *Mucispirillum* showed a significant positive correlation ($P < 0.01$) with fumaric

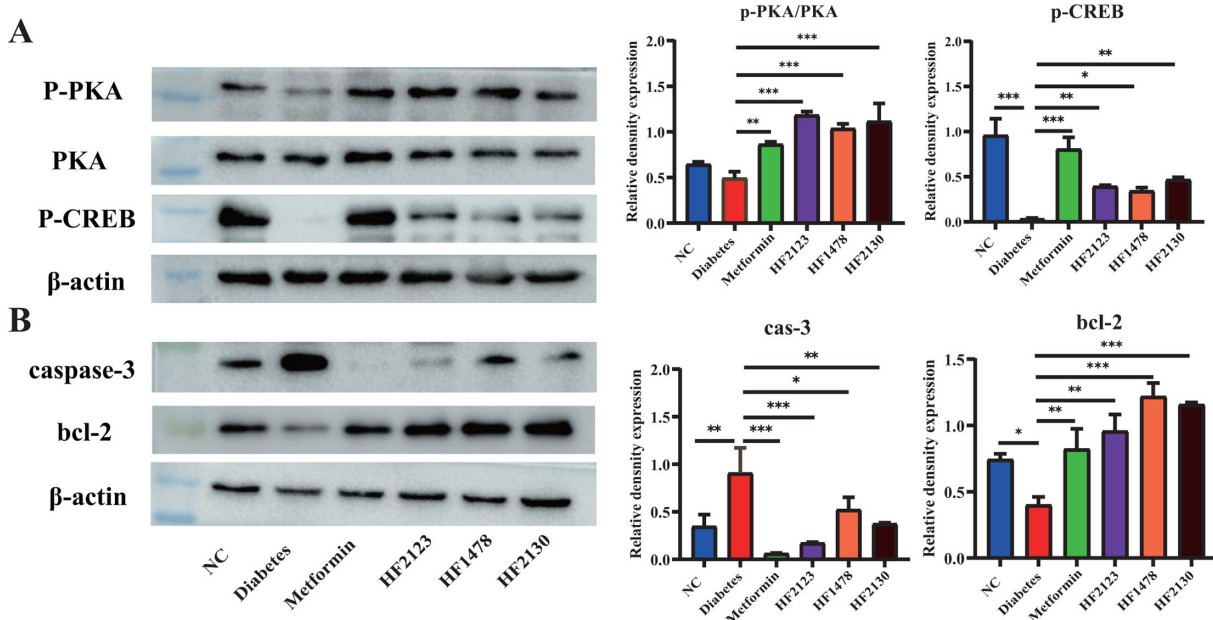

**FIG 5** *P. copri* upregulates cAMP/PKA signaling pathway. (A) The protein levels of PKA, p-PKA, and p-CREB in the colon were detected by the western blot. (B) The protein levels of p-Caspase-3 and p-Bcl-2 in pancreas tissues were detected by the western blot. Gray scale values were analyzed by Image J. Data are expressed as mean ± SD values. *$P < 0.05$, **$P < 0.01$, ***$P < 0.001$.

acid. Changes in bilirubin showed a significant negative correlation ($P < 0.05$) with the abundance of *Akkermansia* and a significant positive correlation ($P < 0.05$) with the abundance of *Mucispirillum*. Glutamine levels were significantly positively correlated ($P < 0.05$ or $P < 0.01$) with the abundance of *Bacteroides*, *Faecalibacterium*, *Prevotella,* and *Akkermansia*, and significantly negatively correlated ($P < 0.05$ or $P < 0.01$) with the *Lachnospiraceae* NK4A136 group and *Mucispirillum*. Additionally, the increased insulin was significantly positively correlated with the abundance of *Bacteroides* ($P < 0.05$), and the increased GLP-1 was significantly positively correlated with the abundance of *Faecalibacterium* and *Prevotella* ($P < 0.05$ or $P < 0.01$).

## Differential gene analysis

To explore the gene content differences between strain HF2130 and the three other strains (HF2123, HF1478, and DSM 18205), we found that 1,037 genes were specific to HF2130, including 669 hypothetical proteins and 327 known genes (Table S4). There is a relatively high abundance of genes encoding glycoside hydrolases (GH109, GH2, GH20, GH29, and GH92) and glycosidyltransferases (GT2) among the genes specific for HF2130 (Fig. S4). The functions of the unique genes were predicted by comparison to the Clusters of Orthologous Groups of proteins (COG) database and were mainly related to cell wall/membrane/envelope biogenesis, replication, recombination, and repair; coenzyme transport and metabolism; and carbohydrate transport and metabolism (Fig. S4).

## DISCUSSION

Although some studies have demonstrated the relationship between *P. copri* and glucose metabolism, contradictory results have been obtained in animal experiments (12, 13), and the efficacy of *P. copri* supplementation in alleviating hyperglycemia and the potential mechanisms remain unclear. In this study, we first isolated 39 *P. copri* strains to obtain a better representation of the species diversity for the subsequent experiments and then examined the effects of supplementation with three live *P. copri* strains in db/db mice. We found that *P. copri* regulated the gut microbiota and metabolism as key mechanisms underlying the beneficial role of *P. copri* in antihyperglycemia.

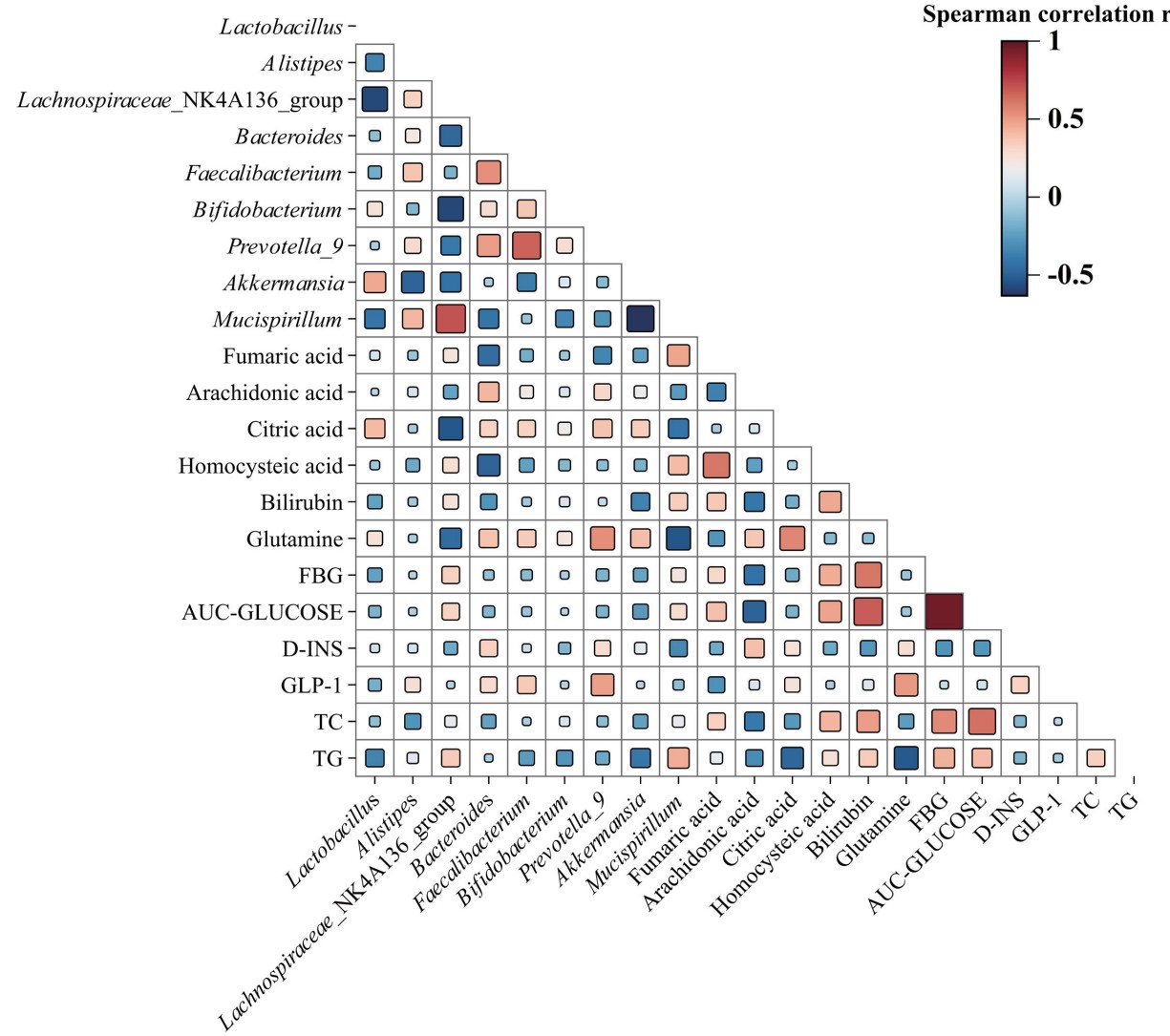

**FIG 6** Spearman's correlation analysis of gut microbiota with differential metabolic profiles. Red squares are positive correlation, and blue squares are negative correlation; the darker the color, the more the significance.

Many studies have associated gut microbiota with the development of T2DM (14, 15). In this study, 16 s RNA fecal microbiota analysis showed that *P. copri* improved the gut microflora imbalance in db/db mice, with an increase in Bacteroidetes and a decrease in Firmicutes. This result is usually observed in healthy individuals without insulin resistance (16). At the genus level, we observed a significant increase in the abundance of *P. copri* after 4 weeks of *P. copri* intervention, indicating successful colonization of *P. copri* in the intestine. Conversely, the abundance of *P. copri* was decreased in the feces of the diabetes group. Similarly, decreased levels of the genus *Prevotella* have been observed in other mice models of diabetes mellitus (17, 18). However, there are two distinct results on *Prevotella* abundance in clinical studies of T2DM patients. Some studies found that the abundance of *Prevotella* was increased in T2DM patients (19, 20), whereas others reported that the abundance of *Prevotella* was significantly reduced in T2DM patients compared with healthy participants (21, 22). This discrepancy may be due to different enterotypes between patients and controls. The human gut microbiome has three enterotypes. *Prevotella* is one of them and is negatively correlated with the abundance of *Bacteroides* (1, 4). However, most studies have not categorized patients according to enterotype, which may bias the results on the effect of different enterotypes on patients. One study classified T2DM patients based on the enterotypes and matched healthy

individuals with the same enterotypes as controls (23). *Prevotella* abundance was decreased in T2DM patients, which is consistent with our results. Our study emphasizes the importance of the gut microbiota in the development of diabetes as in other similar studies. Further studies could explore the potential association between different enterotypes and T2DM patients, and how this relationship affects the disease process and possible interventions.

In addition, we focused on the changes in the abundance of bacteria, which have been shown to be associated with ameliorating insulin resistance in previous studies (24–26). Our finding indicated a significant increase in the abundance of *Akkermansia*, *Faecalibacterium,* and *Bacteroides* following intervention with *P. copri*. *Akkermansia* colonizes the intestinal mucosa and improves glucose homeostasis in mice through the secretion of a GLP-1-inducing protein (27). In clinical studies, patients with T2DM have shown improved glucose metabolism with *Akkermansia* supplementation (28, 29). Another study reported that *Faecalibacterium* may serve as a biomarker for T2DM due to its reduced abundance in T2DM patients (30). Additionally, the intestinal barrier and ZO-1 expression can be restored by the anti-inflammatory molecules of *Faecalibacterium prausnitzii* under diabetic conditions (31). In addition, we found a positive correlation between the elevated *Faecalibacterium* abundance and the increase in GLP-1. Notably, insulin resistance is considered to be negatively correlated with the abundance of *Bacteroides*, which preferentially consume glucose, mannose, and glucosamine to drive the production of their metabolites (32). In this study, we also found a significant positive correlation between increased insulin and elevated *Bacteroides* abundance. In contrast, *Lachnospiraceae* is positively correlated with insulin resistance, and its abundance could be helpful in detecting diabetes (32, 33). These findings further highlight that *P. copri* may promote insulin or GLP-1 secretion by modulating the gut microbiota in db/db mice.

Evidence for the association between microbial metabolites and T2DM is crucial, and a growing number of studies have suggested that changes in the species and abundance of gut microbiota are associated with metabolism disorders and may influence diabetes (34, 35). There are currently two different views on the effects of *P. copri* on glucose metabolism. Pedersen *et al*. found increased glucose intolerance, elevated serum total BCAA levels, and reduced insulin sensitivity in mice when fed a high-fat diet together with the *P. copri* strain DSM 18205 for 3 weeks (13). However, Kovatcheva-Datchary *et al*. found that *P. copri* improved glucose tolerance and increased succinate concentrations after mice were gavaged with *P. copri* for 7 days, thereby improving glucose tolerance and insulin sensitivity (36). We found that there were differences in the treatment of the mouse experiments in these two studies. The first study used the high-fat diet to mimic the environment in which human T2DM develops, whereas the latter used a standard diet. There is a significant difference in fiber content between these two diets, since *P. copri* mainly utilizes plant polysaccharides for energy and far less fat and protein (37, 38). Therefore, differences in the fiber content of the mice's diets would directly affect the growth and activity of *P. copri* in the gut, which in turn may lead to differences in experimental results. This difference in experimental design makes it difficult to directly compare the results of the studies, thus increasing the uncertainty of the conclusions. In this study, we also used the standard diet, and our findings are in agreement with Kovatcheva-Datchary (36) that *P. copri* could improve glucose tolerance.

In this study, fumaric acid and homocysteine were significantly reduced, whereas glutamine was elevated in the *P. copri* treatment groups, as revealed by fecal metabolomics analysis. Some studies on diabetes and its complications have suggested that the accumulation of fumaric acid leads to the overproduction of free radicals, which in turn leads to oxidative stress, thus aggravating the diabetic condition (39, 40). Insulin resistance leads to elevated homocysteine concentrations (41, 42). The abundance of Bacteroides showed a significant negative correlation with fumaric acid and homocysteine levels in this study. In addition, the increase in glutamine may improve impaired glucose tolerance in T2DM by modulating GLP-1 secretion (43). This is consistent with

our findings, and a significant positive correlation was also found between glutamine levels and the abundance of *Bacteroides*, *Faecalibacterium*, *Prevotella*, and *Akkermansia*. Notably the cAMP signaling pathway was enriched in the HF2130 treatment group. Intracellular cAMP is generated by the activation of GPCR in the colon, which in turn activates the key downstream target protein PKA (44). The activation of PKA increases the expression of the *GCG* gene and generates inactive GLP-1, which is then cleaved by *PC1/3* to generate active GLP-1 to enhance insulin secretion and inhibit glucagon secretion, leading to the lowering of blood glucose (45). Furthermore, PKA upregulates the expression of CREB-related genes and reduces fasting blood glucose by inhibiting gluconeogenesis (46). Our results are consistent with these findings. We confirmed the enhanced expression of p-PKA and p-CREB proteins in the colonic tissues of db/db mice treated with *P. copri*. In addition, mRNA levels of *GCG* and *PC1/3*, key genes for GLP-1 sgaiynthesis and secretion, were elevated in colonic tissues and NCI-H716 cells. These results suggest that *P. copri* may upregulate the intestinal cAMP/PKA signaling pathway to promote GLP-1 secretion through its own metabolism or it may affect the gut metabolites by influencing the changes of gut microbiota and achieve the effect of reducing blood glucose through the combined action. The factors that affect the gut microbiota and various metabolites are complex, and further research is needed to confirm their mechanism of action.

Moreover, HF2130 has more copies of genes from the GH29 and GH95 families. The alpha-L-fucosidase of these glycoside hydrolase families is involved in the synthesis of fucosyl-N-acetylglucosamin disaccharides (47), which may enhance the abundance of *Lactobacillus* and *Bifidobacterium* species and inhibit the adhesion of enteropathogenic *Escherichia coli* (48).

Our study has some limitations. First, the sample size of our study was limited, and only one dose of strain intervention was used. More studies are needed to expand the sample size and evaluate the effects of different doses on diabetic mice. Besides, we only conducted a 4-week experiment on one type of diabetic model mice and only studied mice of one gender, which limited the universality of our results. To fully validate our findings, future studies need to use more diverse animal models and extend the duration of the experiment to assess the effects of long-term *P. copri* supplementation on diabetes progression. This includes evaluating the effect of *P. copri* on the gut microbiota by comparing changes in the gut microbiota at different time points. In addition, although *P. copri* is a resident bacterium in healthy individuals, it is currently not listed in the food-grade directory. Therefore, before advancing to clinical trials, we need to ensure that it will not cause any harmful effects on the host, including assessing potential adverse reactions, toxicity, and interactions with other drugs. We will also need to evaluate its stability, viability, and functional capacity during storage from production to market.

## Conclusion

In conclusion, *P. copri* can ameliorate hyperglycemia in db/db mice. The mechanism of action may be related to the improvement of gut microbiota homeostasis, regulation of amino acid metabolism, and activation of cAMP signaling pathway (Fig. 7). All three *P. copri* strains tested alleviated the abnormal glucose metabolism in db/db mice to different degrees, among which strain HF2130 showed better results in various indexes and may be a candidate as a potential probiotic to reduce hyperglycemia.

## MATERIALS AND METHODS

### Bacterial isolation, genome sequencing, and phylogenetic analyses

Fecal sample collection from human volunteers was reviewed and approved by the Ethics Committee of the National Institute for the Prevention and Control of Infectious Diseases, Chinese Centre for Disease Control and Prevention (Approval No.

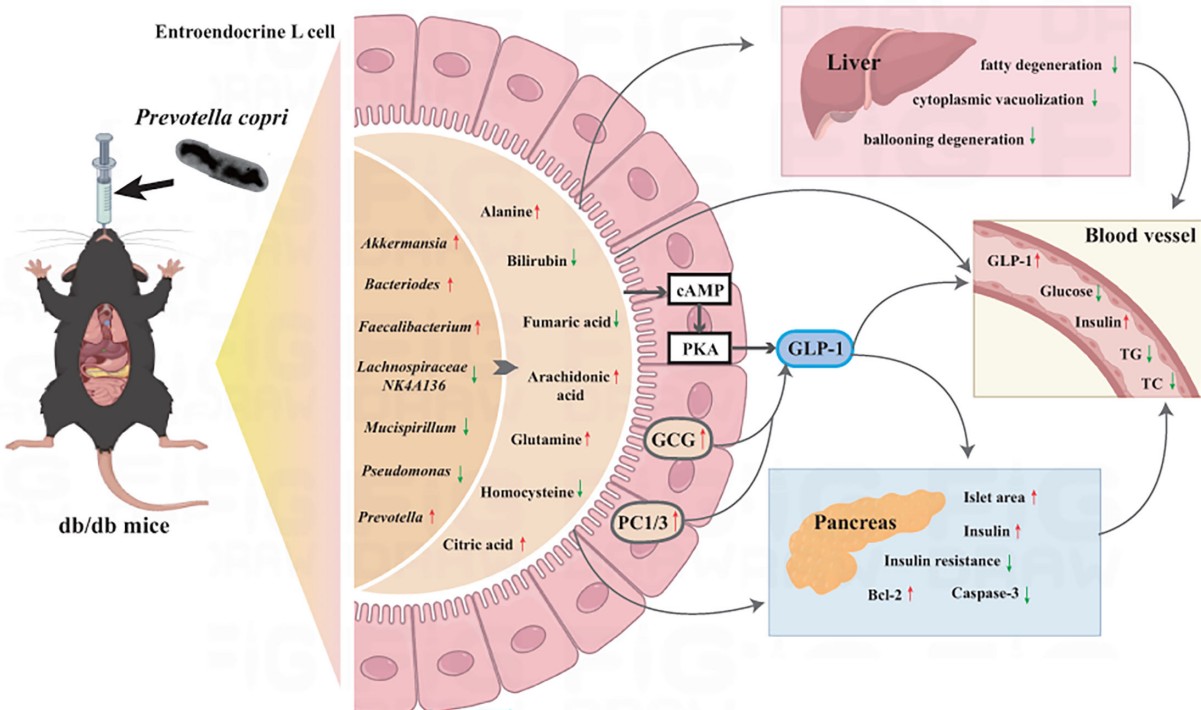

**FIG 7** Schematic representation of the antihyperglycemic effect of *P. copri* on db/db mice.

ICDC-2016007). Fresh healthy human fecal samples were diluted with PBS in an anaerobic chamber and spread onto the culture medium (I, carbon-free medium supplemented with 5% xylan; II, carbon-free medium supplemented with 5% inulin; III, brain heart infusion (BHI) agar containing 5% defibrinated sheep's blood). After incubating for 48–72 h at 37°C, single colonies (colonies with yellow rings on medium I and II) were picked and spread onto new plates to obtain pure cultures. *P. copri*-specific primers (Table S1) were used for PCR amplification (94°C for 5 min, 30 cycles of 94°C, 30 s; 55°C, 30 s; and 72°C, 90 s; and extension at 72°C, 8 min), and *P. copri* DSM 18205$^T$ was used as a positive control. The positive isolates were initially characterized by 16S rRNA gene Sanger sequencing and blast searches in the rRNA/ITS database of the National Center for Biotechnology Information (NCBI). Genomic DNA was extracted using the Wizard Genomic DNA Purification Kit (Promega). Then, 150 bp pair-end sequencing library was generated for genomic sequencing using the Illumina NovaSeq PE150 platform. Genomes were assembled using SOAPdenovo v2.04 (https://bio.tools/soapdenovo) (49). The public genome sequences identified as *P. copri* were obtained from the NCBI GenBank database on 1 May 2022. The sequences were submitted to Quast V5.0.2 (50) for quality control with the criteria of N50 values of >11,000 bp, <500 undetermined bases per 100,000 bases, and the length size in 3Mb ~5 Mb, after removing the duplicate sequences (Fig. S1A; Table S2). A phylogenetic tree was constructed using high-quality sequences screened from public databases and the genome sequences of the strains isolated in this study. All genomic feature files were predicted using Prodigal v2.6.3 (51). Core gene alignment of those *P. copri* strains was estimated using Roary v3.13 (52) for phylogenetic analysis. Finally, core gene phylogenetic trees were constructed using FastTree (53), rooted using *Prevotella melaninogenica* ATCC 25845$^T$ genome as outgroup and displayed using Dendroscope3 (54). An unrooted evolutionary tree within clade A was constructed from 928 core genes extracted from 158 genomes by Roary. Calculation of the ANI across the genome was done using pyani (https://github.com/widdowquinn/pyani) (55), and the matrix of ANI values for the genome was visualized through the R package pheatmap.

## Genomics analysis

Antibiotic resistance genes were predicted using AMRFinderPlus (56). The three strains were tested for penicillin G, ampicillin, clindamycin, clarithromycin, erythromycin, and azithromycin according to the micro broth dilution method developed by Clinicla and Laboratory Standards Institute (CLSI) (57). *Bacteroides thetaiotaomicron* ATCC 29741 was used as quality control strain. The virulence genes were predicted using the VFDB with the identity cutoff at 80% (58). The differences in functional genes were analyzed using BLASTp (BLAST +2.12.0), and the results were visualized using GCview. Carbohydrate-active enzymes (CAZymes) were identified using dbCAN2 (https://github.com/linnab-rown/run_dbcan) (59). Differential genes were annotated using the COG database (60).

## Glucose consumption determination

Pellets of $5*10^6$ CFU *P. copri* were resuspended in 1 mLRPMI-1640 medium (Gibco) and cultured for 24 h; the concentration of glucose in the medium was detected using the glucose oxidase assay (Applygen, China).

## Strains preparation, animals, and treatments

The test strains were incubated in BHI broth for 24 h and then centrifuged at $10,000 \times g$ for 10 min. Then, we washed them twice with sterile phosphate-buffered saline (PBS) and resuspended. All collected samples were diluted to reach $10^8$ CFU/mL. The animal research protocol was approved by the Animal Ethics Committee of the National Institute for Communicable Disease Control and Prevention at the Chinese Center for Disease Control and Prevention (Approval No. IACUC2022-20). In this study, we used the C57BL/KsJ-db/db diabetes mouse model, which is a spontaneous diabetes mouse model with a stable genetic background, which has been used in many studies (61, 62). The dose used was $10^8$ CFU which was based on a previous study using *P. copri* strain DSM 18205 (36). Six-week-old female specific pathogen-free (SPF) grade C57BL/KsJ-db/db (30–40 g) and C57BL/6J (<20 g) mice were purchased from Sipeifu (Beijing) Biotech Co. Mice were housed at 22°C ± 2°C and 50% ± 5% humidity under a 12-h light/dark cycle with regular feed and water. After 1 week of acclimatization, C57BL/6 J mice as normal control (NC) were given PBS (0.2 mL/day) orally. Diabetic db/db mice were randomly divided into five groups of six mice each: diabetes group, orally treated with PBS (0.2 mL/day); metformin group, orally treated with metformin (0.3 g/kg/day); and the three *P. copri* treatment groups (HF2123, HF1478, and HF2130) orally treated with bacterial strains HF2123, HF1478, and HF2130, respectively, with the same bacterial dose ($10^8$ CFU/day). All experimental groups were gavaged at 15:00 every day for 4 weeks, and the body weights of the mice were recorded every 2 days.

## Mammalian cell culture

The human enteroendocrine L cell line NCI-H716 was cultured in low-glucose Dulbecco's Modified Eagle's Medium (DMEM; Gibco) including 10% fetal bovine serum (FBS) at 37°C with 5% $CO_2$. The cells were maintained in matrigel with high-glucose DMEM and 10% FBS for 2 days to mature into endocrine cells. Penicillin (100 U/mL) and streptomycin (100 mg/mL) were added to the culture media.

## Fasting blood glucose and oral glucose tolerance test

Mice were fasted for 8 h, and then tested for fasting blood glucose (FBG) by disinfecting the tail with an alcohol sponge, taking blood from the tail vein, throwing away the first drop of blood, and then testing the second drop with a glucometer. FBG was tested once a week. After 4 weeks of intervention, all mice were starved for 8 h, and an oral glucose tolerance test (OGTT) was performed with 2 g/kg body weight of glucose. Blood glucose values were measured with a glucometer at 0, 30, 60, 90, and 120 min of oral glucose

administration. The area under the curve (AUC) of the glucose was calculated for the corresponding periods.

## Determination of insulin and GLP-1

The mice were fasted for 8 h and blood was collected from the tail vein at 0, 15, and 30 min. The blood was placed in a centrifuge tube with heparin and allowed to stand for 1 h at 4℃, followed by centrifugation at 2,000 × $g$ for 20 min at 4℃. The upper plasma layer was transferred to a clean tube for enzyme-linked immunoassay, in which plasma insulin and GLP-1 levels were measured by ELISA according to the instructions. The insulin resistance index was determined by multiplying the AUC of blood glucose (0–120 min) and plasma insulin (0 and 30 min) obtained after an oral glucose tolerance test.

## Biochemical and histological analysis

The mice were euthanized by removing their eyeballs and performing cervical dislocation. Blood samples were then collected and stored in clean centrifuge tubes. Serum was obtained by centrifugation (2,000 × $g$, 20 min) and frozen at −80℃ until biochemical analysis. Serum total cholesterol (TC), triglyceride (TG), high-density lipoprotein cholesterol (HDL-C), and low-density lipoprotein cholesterol (LDL-C) were determined by ELISA kits according to the instructions. The pancreas, liver, and colon were collected and fixed overnight in 4% paraformaldehyde, embedded in paraffin, and cut into 5-µm sections. Slides were stained with hematoxylin and eosin and then observed with a biomicroscope.

## Gut microbial analysis

Stool samples were collected and stored at −80℃. The DNA extraction of these frozen stool samples was performed according to the protocol of the QIAamp PowerFecal DNA Kit (Germany, QIAGEN). The abundance of *P. copri* before and after the intervention was quantified by qPCR. In addition, the V3–V4 regions of the 16S rRNA gene were amplified and sequenced using the Illumina NovaSeq PE250 platform. The raw data were initially processed and filtered using USEARCH v10.0 to yield clean data, which were then used to identify Amplicon Sequence Varian (ASV) through noise reduction using DADA2 (63). The classify-sklearn algorithm of QIIME2 was applied to annotate species using sliva138 database for each ASV using a pre-trained Naive Bayes classifier (64). LEfSe was used to achieve downscaling and assess the effect of significantly different species. The differences in genus level between groups were compared by one-way analysis of variance (ANOVA) and Turkish post-hoc test.

## Quantitative real-time PCR

Mature NCI-H716 cells were cultured in high-glucose Dulbecco's Modified Eagle's Medium (DMEM; Gibco) and collected after 24 h of intervention with different treatments. Colon samples from all mice were collected into tubes containing RNAlater and stored at −80℃. Total RNA was extracted from cells or colon tissue using Trizol reagent (Ambion, USA). Then, the extracted RNA was reverse-transcribed using the PrimeScript RT Master kit (Takara, Tokyo, Japan) to generate complementary DNA (cDNA). Reverse transcription was performed at 37℃ for 15 min and then 85℃ for 5 min. Quantitative RT-PCR was performed with TB Green Premix Ex Taq GC (Takara, Tokyo, Japan) with three replicates per sample. PCR reaction conditions were as follows: 95℃ for 30 s for initial denaturation; 95℃ for 10 s and 60℃ for 30 s for 40 cycles, followed by dissociation curve analysis. The relative mRNA expression of the genes of interest was calculated by the $2^{-\Delta\Delta CT}$ method and normalized to the expression of β-actin. The primers used for qPCR are listed in Table S1.

## Nontargeted metabolomics by LC-MS analysis

Fecal samples were taken as 100 mg in 80% methanol, then vortexed and ice-bathed for 5 min. After centrifugation at $15,000 \times g$ for 20 min at 4°C, the supernatant was diluted with mass spectrometry-grade water to a methanol content of 53%. Centrifugation was performed again at $15,000 \times g$ for 20 min at 4°C, and the supernatant was collected and analyzed by Q Exactive HF (Germany) and Vanquish UHPLC (Germany). The parameters of the tests were the same as those described previously for serum metabolite assays. The obtained metabolites and relative quantification results were converted by the software metaX and then subjected to PLS-DA of the data to derive Variable Importance in the Projection (VIP) values for each metabolite. Bubble plots were performed with the R package ggplot2, and the KEGG database was used to investigate the function and metabolic pathways of the metabolites.

## Western blot analysis

The colon and pancreas samples were collected and lysed in RIPA lysis buffer containing a 1% protease inhibitor cocktail and a 1% phosphatase inhibitor cocktail. The protein was extracted from the supernatant by centrifugation at 4°C and $10,000 \times g$ for 20 min and measured for protein concentration. Total protein was separated by SDS-PAGE and transferred to the polyvinylidene fluoride (PVDF) membrane. Subsequently, the membranes were incubated overnight with monoclonal mouse anti-$\beta$-actin (1:4000, TransGen, China), anti-caspase-3 (1:1000, CST, USA), anti-Bcl-2 (1:2000, Abcam, UK), anti-p-PKA (1:1000, CST, USA), anti-PKA (1:1000, CST, USA), and anti-p-CREB (1:5000, Abcam, UK). Following this, the membrane was incubated with horseradish peroxidase (HRP)-conjugated goat anti-rabbit IgG (1:1000, Beyotime, China). Finally, the strips were visualized using Amersham Hyperfilm ECL and MP Autoradiography film (GE Healthcare).

## Statistical analysis

Data were reported as mean ± standard error. The data were analyzed using statistical product and service solutions (SPSS) 26.0 software. Differences between multiple groups were compared by one-way ANOVA with Turkey's post-hoc tests or the Kruskal-Wallis test with Dunn's test. The grayscale values of the protein bands were analyzed by Image J software. Significant differences were expressed as $P < 0.05$.

### ACKNOWLEDGMENTS

This work was supported by grants from the Research Units of Discovery of Unknown Bacteria and Function (2018RU010).

C.Y.: Data curation, Formal analysis, Visualization, and Writing-original draft; R.L.: Supervision; Validation, and Writing-original draft; L.Z.: Investigation and Methodology; J.P.: Resources, Software, and Visualization; D.H.: Methodology and Visualization; J.Y.: Conceptualization nad Project administration; H.Z.: Investigation and Methodology; L.H., L.Y., and D.J.: Methodology; L.L. and J.X.: Conceptualization, Supervision, Funding acquisition, and Writing - review and editing

### AUTHOR AFFILIATIONS

[1]Department of Epidemiology, School of Public Health, Shanxi Medical University, Taiyuan, Shanxi, China
[2]National Key Laboratory of Intelligent Tracking and Forecasting for Infectious Diseases, National Institute for Communicable Disease Control and Prevention, Chinese Center for Disease Control and Prevention, Beijing, China
[3]Key Laboratory of Coal Environmental Pathogenicity and Prevention, Shanxi Medical University, Taiyuan, China
[4]School of Biotechnology and Biomolecular Sciences, University of New South Wales, Sydney, New South Wales, Australia

⁵Research Units of Discovery of Unknown Bacteria and Function, Chinese Academy of Medical Sciences, Beijing, China
⁶Hebei Key Laboratory of Intractable Pathogens, Shijiazhuang Center for Disease Control and Prevention, Shijiazhuang, Hebei, China
⁷Institute of Public Health, Nankai University, Tianjin, China

## AUTHOR ORCIDs

Caixin Yang http://orcid.org/0009-0007-0230-3215
Ruiting Lan http://orcid.org/0000-0001-9834-5258
Dong Jin http://orcid.org/0000-0001-9701-7674
Liyun Liu http://orcid.org/0000-0003-2257-0277

## FUNDING

| Funder | Grant(s) | Author(s) |
|---|---|---|
| Research Unit of Discovery of Unknown Bacteria and Function | 2018RU010 | Liyun Liu |

## DATA AVAILABILITY

The 16S rRNA amplicon sequencing data used in this study are publicly available at the NCBI Sequence Read Archive (SRA) database (Bioproject number: PRJNA1043404). The genome sequences of strains isolated in this study are publicly available at the NCBI nucleotide Whole Genome Shotgun (WGS) database (Bioproject number: PRJNA792012 and PRJNA792428).

## ADDITIONAL FILES

The following material is available online.

### Supplemental Material

**Fig. S1 (mSystems00532-24-s0001.eps).** Phylogenetic analysis of *P. copri*.
**Fig. S2 (mSystems00532-24-s0002.eps).** Effect of *P. copri* on glucose consumption.
**Fig. S3 (mSystems00532-24-s0003.eps).** Effect of *P. copri* on serum TC, TG,HDL-C, and LDL-C.
**Fig. S4 (mSystems00532-24-s0004.eps).** Differential distribution of functional gene categories in *P. copri* HF2130, DSM 18205, HF2123, and HF1478.
**Supplemental material (mSystems00532-24-s0005.docx).** Legends for Fig. S1-S4; Tables S1-S4.

### Open Peer Review

**PEER REVIEW HISTORY (review-history.pdf).** An accounting of the reviewer comments and feedback.

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
