## [Reviewer comments · mSystems]

***Prevotella copri* alleviates hyperglycemia and regulates gut microbiota and metabolic profiles in mice**

Caixin Yang, Ruiting Lan, Lijun Zhao, Ji Pu, Dalong Hu, Jing Yang, Huimin Zhou, Lichao Han, Lin Ye, Dong Jin, Jianguo Xu, and Liyun Liu

Corresponding Author(s): Liyun Liu, National Institute for Communicable Disease Control and Prevention

Review Timeline:

Submission Date:

May 9, 2024

Accepted:

June 2, 2024

Editor: Hongwei Zhou

Reviewer(s): Disclosure of reviewer identity is with reference to reviewer comments included in decision letter(s). The following individuals involved in review of your submission have agreed to reveal their identity: Zhuang Li (Reviewer #1); Ruifu Yang (Reviewer #3)

Transaction Report:

DOI: <https://doi.org/10.1128/msystems.00532-24>

Re: mSystems00532-24 (*Prevotella copri* alleviates hyperglycemia and regulates gut microbiota and metabolic profiles in mice)

Dear Dr. Liyun Liu:

Your manuscript has been accepted, and I am forwarding it to the ASM production staff for publication. Your paper will first be checked to make sure all elements meet the technical requirements. ASM staff will contact you if anything needs to be revised before copyediting and production can begin. Otherwise, you will be notified when your proofs are ready to be viewed.

Sincerely,
Hongwei Zhou
Editor
mSystems

Reviewer #1 (Comments for the Author):

I have no more comments.

Reviewer #3 (Comments for the Author):

The author's response has answered this reviewer's concerns.